# A clinical score for identifying active tuberculosis while awaiting microbiological results: Development and validation of a multivariable prediction model in sub-Saharan Africa

Yeonsoo Baik[1]*, Hannah M. Rickman[1], Colleen F. Hanrahan[1], Lesego Mmolawa[2], Peter J. Kitonsa[3], Tsundzukana Sewelana[2], Annet Nalutaaya[3], Emily A. Kendall[3,4], Limakatso Lebina[2], Neil Martinson[2], Achilles Katamba[3,5], David W. Dowdy[1,3]

1 Department of Epidemiology, Johns Hopkins Bloomberg School of Public Health, Baltimore, Maryland, United States of America, 2 Perinatal HIV Research Unit, Nurses Residence, Chris Hani Baragwanath Academic Hospital, Soweto, South Africa, 3 Uganda Tuberculosis Implementation Research Consortium, Makerere University, Kampala, Uganda, 4 Division of Infectious Diseases, Department of Medicine, Johns Hopkins University, Baltimore, Maryland, United States of America, 5 Clinical Epidemiology and Biostatistics Unit, Department of Medicine, Makerere University College of Health Sciences, Kampala, Uganda

* ybaik3@jhmi.edu

## Abstract

### Background

In highly resource-limited settings, many clinics lack same-day microbiological testing for active tuberculosis (TB). In these contexts, risk of pretreatment loss to follow-up is high, and a simple, easy-to-use clinical risk score could be useful.

### Methods and findings

We analyzed data from adults tested for TB with Xpert MTB/RIF across 28 primary health clinics in rural South Africa (between July 2016 and January 2018). We used least absolute shrinkage and selection operator regression to identify characteristics associated with Xpert-confirmed TB and converted coefficients into a simple score. We assessed discrimination using receiver operating characteristic (ROC) curves, calibration using Cox linear logistic regression, and clinical utility using decision curves. We validated the score externally in a population of adults tested for TB across 4 primary health clinics in urban Uganda (between May 2018 and December 2019). Model development was repeated de novo with the Ugandan population to compare clinical scores. The South African and Ugandan cohorts included 701 and 106 individuals who tested positive for TB, respectively, and 686 and 281 randomly selected individuals who tested negative. Compared to the Ugandan cohort, the South African cohort was older (41% versus 19% aged 45 years or older), had similar breakdown of biological sex (48% versus 50% female), and had higher HIV prevalence (45% versus 34%). The final prediction model, scored from 0 to 10, included 6 characteristics: age, sex, HIV (2 points), diabetes, number of classical TB symptoms (cough, fever, weight loss,

**Data Availability Statement:** The data are held in the Johns Hopkins University Data Services database and available at https://doi.org/10.7281/T1/W2AG3A.

**Funding:** The study was funded by the US National Institutes of Health (R01AI116787 and R01HL138728) to DWD. https://www.nih.gov/grants-funding The funders had no role in the study design, data collection and analysis, decision to publish, or preparation of the manuscript.

**Competing interests:** I have read the journal's policy and the authors of this manuscript have the following competing interests: "NM's institution receives a grant from Pfizer to follow up patients with pneumonia."

**Abbreviations:** ART, antiretroviral therapy; BMI, body mass index; IQR, interquartile range; IRB, Institutional Review Board; ROC, receiver operating characteristic; TB, tuberculosis; TRIPOD, Transparent Reporting of a multivariable prediction model for Individual Prognosis Or Diagnosis; WHO, World Health Organization.

and night sweats; 1 point each), and >14-day symptom duration. Discrimination was moderate in the derivation (c-statistic = 0.82, 95% CI = 0.81 to 0.82) and validation (c-statistic = 0.75, 95% CI = 0.69 to 0.80) populations. A patient with 10% pretest probability of TB would have a posttest probability of 4% with a score of 3/10 versus 43% with a score of 7/10. The de novo Ugandan model contained similar characteristics and performed equally well. Our study may be subject to spectrum bias as we only included a random sample of people without TB from each cohort. This score is only meant to guide management while awaiting microbiological results, not intended as a community-based triage test (i.e., to identify individuals who should receive further testing).

## Conclusions

In this study, we observed that a simple clinical risk score reasonably distinguished individuals with and without TB among those submitting sputum for diagnosis. Subject to prospective validation, this score might be useful in settings with constrained diagnostic resources where concern for pretreatment loss to follow-up is high.

## Author summary

### Why was this study done?

- In high-burden settings, up to 40% of patients with confirmed tuberculosis (TB) are lost to follow-up between the time they initially present for diagnosis and the time treatment can be initiated.

- A simple clinical prediction rule for initiation of empiric treatment could therefore be very helpful, but existing prediction rules for TB have been designed for use in specific contexts (e.g., HIV clinics and contact investigation), require data and/or infrastructure (e.g., radiography) that are unlikely to be available in highly resource-limited settings, or have been constructed in a format that is difficult to implement for busy midlevel clinicians with little time or access to computers.

### What did the researchers do and find?

- We developed a simple clinical risk score for diagnosis of TB among adults presenting to primary health clinics in rural South Africa and validated the score in urban Uganda.

- This score (ranging from 1 to 10) is easy to calculate by hand and requires only information readily accessible to clinicians in highly resource-limited settings.

- This score adds clinical utility at cutoffs that likely reflect the risk–benefit ratio of making a TB diagnosis when same-day microbiological testing is unavailable.

### What do these findings mean?

- If prospectively validated in other populations, this clinical risk score could improve the process of empiric TB diagnosis in peripheral health settings where access to same-day microbiological testing is lacking and the risk of loss to follow-up is high.

- This clinical risk score might be used for patients at high risk to initiate a short-term empirical course of 4-drug TB treatment on the same day of their clinic visit until their microbiological test results are available.

## Introduction

The World Health Organization (WHO) estimates that 10.0 million new tuberculosis (TB) cases and 1.4 million deaths occurred in 2018, making TB the leading single-agent cause of infectious mortality worldwide [1] Pretreatment loss to follow-up is a major contributor to TB morbidity and transmission: an estimated 13% to 18% of people diagnosed with TB in high-burden settings are lost to follow-up before starting treatment [2]. Rapid treatment of TB is difficult in settings where access to on-site diagnostic testing, radiography, and specialist staff is limited. Empirical (or clinical) TB diagnosis—in other words, diagnosis made without microbiological confirmation—is therefore an important consideration in these settings where the risk of loss to follow-up is high. However, empirical diagnosis is not standardized, may not be routinely made by midlevel clinicians who frequently staff such clinics, and does not correlate well with microbiological results—often leading to inappropriate therapy [3,4]. Simple tools to improve the process of empirical TB diagnosis in severely resource-constrained settings could therefore substantially reduce *Mycobacterium tuberculosis* transmission and TB mortality.

Molecular tests for TB, including Xpert MTB/RIF (Xpert; Cepheid, Sunnyvale, California, United States of America) and Xpert Ultra, have improved TB diagnosis but remain unavailable in many peripheral clinics, despite the availability of newer single-module systems such as Xpert Omni and Xpert Edge [5–7]. Even where available, results from molecular tests are often not received by the treating clinician on the same day. Most randomized trials of Xpert on clinical outcomes have been conducted in settings where same-day testing and ancillary diagnostic procedures (e.g., chest X-ray) were readily available [7–9]. However, a very common clinical scenario is one in which a patient with presumptive TB is unlikely to receive same-day radiological or microbiological test results and is at high risk of loss to follow-up if treatment is not initiated immediately [8,10,11]. In such settings, a simple clinical score that could rapidly identify high-risk patients for consideration of anti-TB treatment would be exceedingly helpful.

Most existing models for predicting active TB require equipment or infrastructure (e.g., radiology, laboratory testing, and calculations requiring a computer or smartphone) that is generally unavailable in clinical settings that also lack same-day microbiological testing for TB [12–15]. Thus, there exists a need for a prediction score that is simple enough to write on paper and calculate rapidly by hand. We aimed to develop such a simple score for predicting active TB, requiring only information likely to be accessible to clinicians in such resource-limited settings. To ensure that this score could be widely applicable across sub-Saharan Africa, we developed the score in 1 epidemiological context (rural South Africa) and validated it in a very different one (urban Uganda).

## Methods

### Study setting and population

We derived our clinical score using data from the Kharitode study, a cluster-randomized controlled trial of TB contact investigation strategies (household and incentive-based contact investigation) against facility-based screening (the standard of care) across 56 primary care clinics in 1 rural and 1 semi-urban district of Limpopo Province, South Africa [16]. Limpopo

Province has a low population density (46.1 people per km$^2$ in 2016) [17], high TB incidence (301 per 100,000/year in 2015), and high adult HIV prevalence (17%) [18]. The median household income in this study population was approximately $150 per month [19].

For the current analysis, we included participants above 15 years of age who presented with TB symptoms at the 28 primary health clinics randomized to facility-based (i.e., passive) screening between July 2016 and January 2018. According to standard practice in South Africa, all clinic attendees were screened for TB symptoms, and symptomatic individuals were referred for Xpert testing. Among the symptomatic individuals, those who provided informed consent were interviewed about demographic and clinical details and provided follow-up contact information [16]. Our definition of active TB was based on an Xpert–positive sputum specimen. Xpert has imperfect sensitivity for pulmonary TB (estimated 88%) but high specificity (98%), such that the positive predictive value of Xpert is expected to be high [20]. Data were collected on all consenting Xpert–positive cases and a random sample of Xpert–negative individuals designed to be representative of the entire Xpert–negative population. Individuals who did not have an interpretable Xpert result or were treated for TB despite a negative Xpert result (<1%) were excluded from analysis.

## Model specification and score generation

This study is reported as per the Transparent Reporting of a multivariable prediction model for Individual Prognosis Or Diagnosis (TRIPOD) guideline developing, validating, or updating a prediction model (S1 Checklist). We considered variables for inclusion in our model that met the following criteria: (1) existing evidence of association with TB disease (i.e., excluding variables measured for reasons other than TB risk); (2) measured in the Kharitode trial with little (<10%) missingness; and (3) likely to be available to clinicians in highly resource-limited settings. Continuous predictors were either categorized based on a priori selection of cutoffs that fell near the midpoint of the observed range (e.g., 2-week duration of symptoms) or handled as ordinal variables (e.g., number of current TB symptoms). We applied multiple imputation with chained equations [21,22] to impute missing data, using the R package "mice." We then conducted 10-fold cross-validation and fit a multivariable least absolute shrinkage and selection operator (lasso) regression model, using the R package "glmnet" [23,24]. Our goal was to create a simple scoring system that could be used rapidly by clinicians in settings of low infrastructure without sacrificing predictive accuracy. To this end, we generated a point scoring system by assigning points to each variable by first identifying clusters of coefficients in the multivariable lasso regression model (i.e., variables that each had roughly equal associations with the outcome of interest), then taking the median of those clustered coefficients, dividing all coefficients by that median value, then rounding to the nearest integer. This process enabled us to remove variables that were only weakly associated with the outcome (i.e., coefficients less than half the median value of the majority of predictors) and to generate a score based entirely on integer values. Use of the mean, rather than median, value did not change our results. Points were summed to generate the final risk score. The final risk score was then fitted to a logistic regression taking TB status (Xpert result) as the outcome. The predictive accuracy of this simplified risk score was compared against that of a full logistic regression model using actual coefficient values (S4 and S5 Figs, S3 Table). We performed a secondary analysis in which all missing values were assigned the most common value (single imputation) to reflect likely clinical use of the score in the field (S4 Table, S8 Fig).

## External validation

We validated the model by assessing its performance in a population of individuals tested for active pulmonary TB in outpatient clinics in densely populated parishes of Kampala, the

capital city of Uganda. The estimated population density in our study site is 23,000 people per km$^2$. Among the adult study population, self-reported median household income was \$91, the prevalence of pulmonary TB was estimated between 420 and 940 per 100,000 population, and the prevalence of HIV was estimated 18%. We enrolled patients with presumptive TB presenting at 4 healthcare facilities in Kampala, Uganda between May 22, 2018 and December 31, 2019. This external validation population consisted of adults (≥15 years) who tested positive for TB on Xpert and a random selection of adults who presented with TB symptoms on the same day as the case but tested negative on Xpert and were not treated for active TB. Cases and non-cases were not matched on any other variable. Missing data were found in only 1 individual who was removed from the analysis. We repeated the same model development process de novo with the bootstrap sample of Ugandan population using a split internal validation approach to assess the degree of difference between the clinical score system as developed in South Africa and the score as developed in Uganda. The de novo analysis was then also validated in the South African population (S5 and S6 Tables, S11 and S12 Figs). All analyses were conducted using R version 3.6.1 (R Foundation for Statistical Computing, Vienna, Austria).

An important aspect of the derivation and validation populations is that a random selection of all non-cases (rather than a full cohort) was enrolled in each study. In South Africa, laboratory registers were used to randomly select 1 Xpert–negative patient for every Xpert–positive patient enrolled; in Uganda, clinical registers of patients with presumptive TB were used to select 2 Xpert–negative patients per Xpert–positive patient. In both studies, controls were matched to cases based only on facility and date range of presentation. Thus, to mitigate against spectrum bias, we performed sensitivity analyses of all outcomes in simulated cohorts where the data from non-cases were replicated to provide prespecified levels of TB prevalence (5%, 10%, and 20%) among people being tested.

## Model calibration, discrimination, and clinical utility

We assessed agreement between model-based risk predictions and observed TB status (calibration) using the Cox linear logistic regression in the validation population [21,25]. This regression technique compares observed TB status (positive versus negative) to the log odds of predicted TB risk, assuming a linear relationship. The intercept of this regression can therefore be interpreted as the overall degree of under- or overestimation of risk, while the slope can be interpreted as evaluating whether the accuracy of risk prediction is different, comparing those with low predicted risk to those with high predicted risk. The intercept of the fitted regression model was modified to reflect the different sampling fraction in each population [26,27]. Discrimination was evaluated using receiver operating characteristic (ROC) curve analysis and calculation of the c-statistic (area under the ROC curve). We conducted decision curve analysis to help weigh the relative benefits of treating patients with true active TB against the harms of treating patients with Xpert–negative results using different cutoffs of the clinical score [28,29]. For both predictive accuracy and decision curve analyses, we weighted data from Xpert–negative individuals in Uganda to represent a scenario of 10% underlying TB prevalence. Finally, to provide a clinically relevant decision tool, we used the sensitivity and specificity of the model to estimate the probability of TB for individuals with each possible risk score (1 to 10) assuming pretest probabilities (i.e., prevalence of TB in the underlying population being tested) of 5%, 10%, and 20%.

## Analysis plan

This analysis, which relied on the availability of overlapping data from 2 independent studies, was not laid out in a prospective protocol. Data to inform the score were determined a priori

based on available variables deemed to be measurable in a highly resource-constrained setting, and the score was developed based on consensus assessment of lasso regression coefficients (i.e., how to combine those coefficients into a simple numerical score), prior to performing the validation. The variables contributing to the score (and the score itself) remain the same as those made in a priori decisions, but numerous supporting analyses were performed or modified in response to comments made in peer review. These revisions included a modified assessment of calibration (use of Cox linear logistic regression with modification of the intercept to reflect sampling fractions), use of multiple imputation for missing data, inclusion of decision curve analysis, and performance of multiple sensitivity analyses (including simulated cohorts to evaluate different underlying TB prevalence, use of restricted cubic splines to model age, and restriction to patients with chronic cough only). Data sufficient to replicate all analyses are available at https://doi.org/10.7281/T1/W2AG3A, and model code is available at https://github.com/ybaik10/clinicalriskscore.

### Ethics statement

The Kharitode study was approved by the University of the Witwatersrand's Human Research Ethics Committee and the Limpopo Provincial Government Department of Health. The external validation study was approved by the Institutional Review Board (IRB) of the Makerere University College of Health Sciences. Both studies were approved by the IRB of the Johns Hopkins Bloomberg School of Public Health through an Authorization Agreement with the local IRBs. All study participants provided written informed consent. Parental consent (with participant assent) was obtained for participants between 15 and 18 years of age.

## Results

A total of 1,387 adults with symptoms of active TB were included in the South African derivation cohort: 701 individuals with Xpert-confirmed TB and 686 individuals who presented to the same clinics were tested for TB with Xpert but tested negative and were not started on TB treatment (Fig 1). The Ugandan external validation population included 106 Xpert–positive and 281 Xpert–negative adults. Participants with TB were more likely to report classic TB symptoms (other than cough) and had a longer duration of symptoms (median 4 weeks versus 2 weeks) compared to people with negative Xpert results in both populations (Table 1). Six of the predictors—male sex, age between 25 and 44 years, HIV positivity (based on clinical

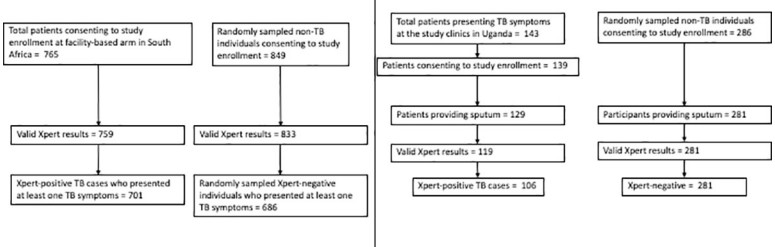

**Fig 1. Flow chart of the South African and Ugandan study population.** Left panel shows the flow chart of the South African study population. Laboratory registers were used to randomly select 1 Xpert–negative patient for every Xpert–positive patient enrolled. After excluding those who did not report any TB symptoms, the final ratio between Xpert–positive TB cases (*n* = 701) and Xpert–negative individuals (*n* = 686) became 1.02:1. Right panel shows the flow chart of the Ugandan study population. Clinical registers of patients with presumptive TB were used to select 2 Xpert–negative patients per Xpert–positive patient. The final ratio of Xpert–positive TB cases (*n* = 106) to Xpert–negative individuals (*n* = 281) was 0.38:1. In both studies, controls were matched to cases based only on facility and date range of clinical presentation. TB, tuberculosis.

**Table 1. Characteristics of model derivation (South Africa) and external validation (Uganda) populations.**

| | South African derivation population, N (%) | | Ugandan external validation population, N (%) | |
|---|---|---|---|---|
| | Xpert–positive, 701 (51) | Xpert–negative, 686 (49) | Xpert–positive, 106 (27) | Xpert–negative, 281 (73) |
| **Age category, years** | | | | |
| 15–24 | 84 (12) | 119 (17) | 20 (19) | 66 (24) |
| 25–34 | 166 (24) | 118 (17) | 42 (40) | 87 (31) |
| 35–44 | 202 (29) | 133 (19) | 30 (28) | 69 (25) |
| 45–54 | 147 (21) | 154 (22) | 14 (13) | 39 (14) |
| ≥55 | 102 (14) | 162 (24) | 0 (0) | 20 (7) |
| **Sex** | | | | |
| Female | 275 (39) | 389 (57) | 38 (36) | 161 (57) |
| Male | 426 (61) | 297 (43) | 68 (64) | 120 (43) |
| **HIV status** | | | | |
| HIV–negative or unknown[a] | 281 (40) | 484 (71) | 68 (64) | 188 (67) |
| HIV–positive, on antiretroviral therapy | 322 (46) | 166 (24) | 26 (25) | 89 (32) |
| HIV–positive, not on antiretroviral therapy | 48 (7) | 14 (2) | 12 (11) | 4 (1) |
| HIV–positive, unknown antiretroviral therapy | 50 (7) | 22 (3) | 0 (0) | 0 (0) |
| **Presence of classical TB symptoms** | | | | |
| Cough | 608 (87) | 653 (95) | 105 (99) | 279 (99) |
| Fever | 305 (44) | 165 (24) | 47 (44) | 67 (24) |
| Weight loss | 431 (62) | 94 (14) | 76 (72) | 95 (34) |
| Night sweats | 414 (59) | 160 (23) | 42 (40) | 45 (16) |
| **Total number of classical TB symptoms** | | | | |
| 1 | 174 (25) | 420 (60) | 20 (19) | 136 (48) |
| 2 | 163 (23) | 173 (25) | 32 (30) | 96 (34) |
| 3 | 198 (28) | 66 (9) | 30 (28) | 38 (14) |
| 4 | 166 (24) | 27 (4) | 24 (23) | 11 (4) |
| **Duration of TB symptoms[b]** | | | | |
| ≤2 weeks | 206 (31) | 378 (55) | 10 (9) | 113 (40) |
| >2 weeks | 464 (69) | 283 (41) | 96 (91) | 168 (60) |
| **Any other non-TB symptoms[c]** | 440 (63) | 301 (44) | 62 (59) | 111 (40) |
| **Self-reported comorbidities** | | | | |
| Diabetes mellitus[b] | 25 (4) | 21 (3) | 1 (1) | 2 (1) |
| Obstructive pulmonary disease[b] | 15 (2) | 21 (3) | 1 (1) | 7 (3) |
| **Previous TB diagnosis (self-report)[b]** | 128 (18) | 86 (13) | 24 (23) | 32 (11) |
| **Education[b]** | | | | |
| High school or less | 511 (73) | 478 (70) | 35 (33) | 96 (34) |
| Any post-high school education | 186 (27) | 200 (29) | 71 (67) | 185 (66) |
| **Smoking history[b]** | | | | |
| Never | 416 (59) | 478 (70) | 67 (63) | 225 (80) |
| Ever | 283 (40) | 200 (29) | 39 (37) | 56 (20) |
| **Occupation[b]** | | | | |
| Regularly employed | 128 (18) | 148 (22) | 58 (55%) | 148 (53%) |
| Irregular work, student, or housewife | 110 (16) | 206 (30) | 31 (29%) | 89 (32%) |
| Unemployed or retired | 458 (66) | 322 (48) | 17 (16%) | 44 (16%) |

(*Continued*)

**Table 1.** (*Continued*)

| | South African derivation population, *N* (%) | | Ugandan external validation population, *N* (%) | |
|---|---|---|---|---|
| | Xpert–positive, 701 (51) | Xpert–negative, 686 (49) | Xpert–positive, 106 (27) | Xpert–negative, 281 (73) |
| Income[b,d] | 1,820 (1,140–3,200) ZAR | 2,125 (1,140–3,500) ZAR | 375,000 (200,000–600,000) Shilling | 340,000 (200,000–600,000) Shilling |

HIV, human immunodeficiency virus; TB, tuberculosis.

[a]A total of 11% of patients whose HIV status was unknown or unreported were included in South African study population.

[b]Missing rate is 4% for duration of TB symptoms; 2% for diabetes and obstructive pulmonary diseases; 1% for education, previous TB, smoking, and occupation; and 43% for income in South Africa.

[c]Includes chest pain, pain elsewhere, skin symptoms, genitourinary symptoms, gastrointestinal symptoms, and "any other symptom" by self-report.

[d]1 ZAR, South African currency = US$0.06; 1 Shilling, Ugandan currency = US$0.0003.

registers or self-reported answers), the number of WHO TB symptoms (cough, fever, and night sweats within the past few days from the day of clinic visit and weight loss of more than 5 kg or enough to make clothes loose) currently experiencing [16], any TB symptoms duration of over 2 weeks, and self-reported history of diabetes—had sufficiently strong associations to be included in the simple clinical score, which ranged from 1 to 10 (Table 2). Fig 1 illustrates the use of this score in settings of different pretest probability (TB prevalence among people

**Table 2. Association of key variables with Xpert-confirmed pulmonary TB.**

| | Unadjusted odds ratio[a] (95% CI) | Adjusted odds ratio[b] (95% CI) | *p*-value | Lasso regression coefficient | Score[c] |
|---|---|---|---|---|---|
| **Age category, years** | | | | | |
| 15–24 | 1.17 (0.81, 1.7) | 1.65 (1.03, 2.65) | 0.04 | 0.39 | |
| 25–34 | 2.3 (1.64, 3.23) | 2.66 (1.72, 4.11) | <0.001 | 0.92 | 1 |
| 35–44 | 2.52 (1.81, 3.5) | 1.77 (1.15, 2.73) | 0.01 | 0.51 | 1 |
| 45–54 | 1.6 (1.15, 2.24) | 1.28 (0.84, 1.97) | 0.25 | 0.21 | |
| ≥55 | Reference | Reference | | Reference | |
| **Sex** | | | | | |
| Female | Reference | Reference | | Reference | |
| Male | 2.03 (1.64, 2.51) | 2.90 (2.07, 4.05) | <0.001 | 0.93 | 1 |
| **HIV status** | | | | | |
| HIV–negative or unknown | Reference | Reference | | Reference | |
| HIV–positive | 3.63 (2.91, 4.53) | 3.55 (2.65, 4.75) | <0.001 | 1.22 | 2 |
| **HIV/antiretroviral therapy status** | | | | | |
| HIV–negative or unknown | Reference | - | | - | |
| HIV–positive, on antiretroviral therapy | 3.38 (2.66, 4.28) | - | | - | |
| HIV–positive, not on antiretroviral therapy[d] | 6.02 (3.26, 11.11) | - | | - | |
| **Classical TB symptoms** | | - | | - | |
| Cough | 0.33 (0.22, 0.5) | - | | - | |
| Fever | 2.44 (1.94, 3.07) | - | | - | |
| Weight loss | 10.08 (7.73, 13.14) | - | | - | |
| Night sweats | 4.74 (3.76, 5.98) | - | | - | |
| **Total number of classical TB symptoms** | | | | | |
| 1 | Reference | Reference | | Reference | 1 (0) |
| 2 | 2.27 (1.72, 3.00) | 1.92 (1.40, 2.64) | <0.001 | 0.69 | 2 (1) |

(*Continued*)

**Table 2.** (Continued)

| | Unadjusted odds ratio[a] (95% CI) | Adjusted odds ratio[b] (95% CI) | *p*-value | Lasso regression coefficient | Score[c] |
|---|---|---|---|---|---|
| 3 | 7.24 (5.21, 10.07) | 5.38 (3.70, 7.83) | <0.001 | 1.70 | 3 (2) |
| 4 | 14.84 (9.52, 23.12) | 10.00 (6.08, 16.46) | <0.001 | 2.31 | 4 (3) |
| **Duration of TB symptoms** | | | | | |
| ≤2 weeks | Reference | Reference | | Reference | |
| >2 weeks | 3.02 (2.41, 3.79) | 2.41 (1.83, 3.16) | <0.001 | 0.85 | 1 |
| **Any other non-TB symptoms[e]** | 2.16 (1.74, 2.68) | 1.35 (1.03, 1.77) | 0.03 | 0 | |
| **Self-reported comorbidities** | | | | | |
| Diabetes mellitus | 1.11 (0.62, 1.99) | 2.02 (0.92, 4.42) | 0.08 | 0.76 | 1 |
| Obstructive pulmonary disease | 0.69 (0.35, 1.35) | - | | - | |
| **Previous TB diagnosis** (self-report) | 1.49 (1.11, 2.00) | 1.18 (0.81, 1.71) | 0.39 | 0.13 | |
| **Education[f]** | | | | | |
| High school or less | Reference | - | | - | |
| Any post-high school education | 0.86 (0.68, 1.08) | - | | - | |
| **Smoking history** | | | | | |
| Never | Reference | Reference | | Reference | |
| Ever | 1.61 (1.29, 2.01) | 0.78 (0.55, 1.11) | 0.16 | −0.23 | 0 |

95% CI, 95% confidence interval; HIV, human immunodeficiency virus; lasso, least absolute shrinkage and selection operator; TB, tuberculosis.

[a]Estimated from univariate logistic regression.

[b]Estimated from the multivariable logistic regression, adjusting for all other variables with a population prevalence of at least 10% and a statistically significant association with TB on univariate regression. Individual TB symptoms were removed in favor of total number of symptoms based on an a priori decision.

[c]To transform the coefficients to simple relative points, each point in this simple clinical score is estimated by dividing the corresponding lasso coefficient by the median value of coefficients (0.9, taking 1 coefficient closer to clustered values when a variable has more than 2 categories) and rounding to the nearest integer. One point was added to the score for number of TB symptoms to increase usability, as all participants had at least 1 symptom.

[d]This category includes HIV–positive with unknown antiretroviral therapy status.

[e]Participants were asked about chest pain, pain elsewhere, skin symptoms, genitourinary symptoms, gastrointestinal symptoms, and "any other symptom."

[f]Occupation and median household income were also explored as indicators of socioeconomic status but excluded based on uncertain applicability in the clinical setting.

being tested). In settings where the pretest probability is 10%, a patient with a risk score of 3 would have a predicted TB risk of 4%, whereas a person with a risk score of 7 would have a predicted risk of 43% (Fig 2). Optimal cutoffs for clinical use were at score of 4 (sensitivity = 0.85, specificity = 0.63) or 5 (sensitivity = 0.69, specificity = 0.80), out of 10 (Fig 3).

In the model derivation and external validation population, the median risk score among people with TB was 5 (interquartile range [IQR] 4 to 6), compared to 3 (IQR 2 to 4) and 4 (IQR = 3 to 5), respectively, in people without TB. Model calibration was acceptable in the context of a very simple score (intercept = −0.25 [95% CI: −0.50, 0.01] and slope = 0.76 [95% CI: 0.65, 0.96]). (Fig 4A) The simplified risk score demonstrated moderate discrimination in the derivation and external validation population, with a c-statistic of 0.82 (95% CI 0.81 to 0.82) and 0.75 (95% CI 0.69 to 0.80), respectively. This was comparable to the discrimination of the full model using regression coefficients (c-statistic 0.83, 95% CI 0.82 to 0.83 in the derivation population and 0.77, 95% CI 0.71 to 0.82 in the validation population) (Figs 4B and S4). Estimated TB prevalence and mode of imputing missing data did not influence model discrimination (S8 Fig).

Decision curve analysis suggested that use of the clinical score to inform empiric treatment decisions offered a positive net benefit (compared to a treat-all or treat-none strategy) when the ratio of benefit of a true-positive diagnosis to cost of a false-positive diagnosis ranged from 4:1 to 50:1 (Fig 5).

**Step 1. Calculate Risk Score**

**Step 2. Calculate Your Patient's Probability of TB**

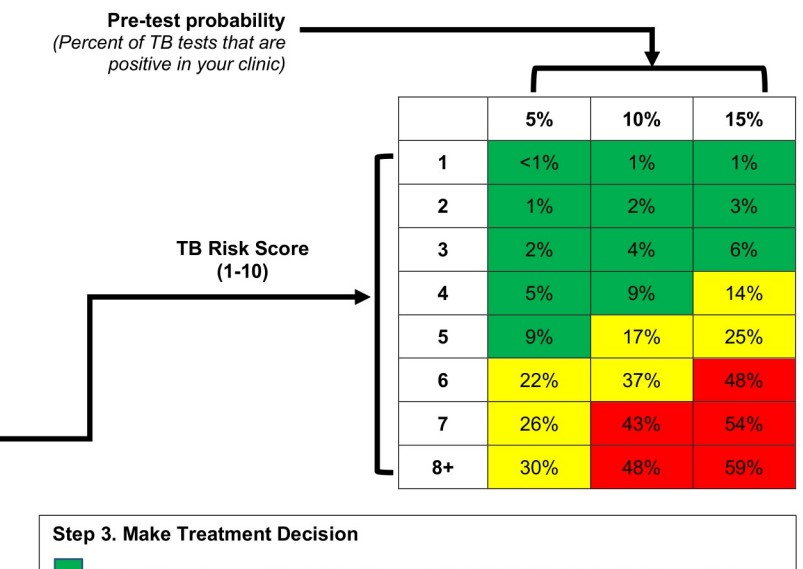

| Risk factor | Points |
|---|---|
| Male sex | □/1 |
| Age 25 – 44 years | □/1 |
| HIV positive | □/2 |
| Diabetes* | □/1 |
| Symptom duration >14 days* | □/1 |
| Number of WHO TB symptoms<br> Cough<br> Fever<br> Night sweats<br> Weight loss | □/4 |
| **Total** | □/10 |

If variables are not available in practice, a modal value or the most common value can be used. *In our study, duration of TB symptoms and diabetes mellitus had 4% and 2% missing rate, respectively, and were replaced with 1 and 0.

Pre-test probability
*(Percent of TB tests that are positive in your clinic)*

TB Risk Score
(1-10)

| | 5% | 10% | 15% |
|---|---|---|---|
| 1 | <1% | 1% | 1% |
| 2 | 1% | 2% | 3% |
| 3 | 2% | 4% | 6% |
| 4 | 5% | 9% | 14% |
| 5 | 9% | 17% | 25% |
| 6 | 22% | 37% | 48% |
| 7 | 26% | 43% | 54% |
| 8+ | 30% | 48% | 59% |

**Step 3. Make Treatment Decision**

■ Probability of TB <10% → **DO NOT TREAT WITHOUT A POSITIVE TEST**

■ Probability of TB 10 – 40% → **CONSIDER TREATMENT**

■ Probability of TB ≥ 40% → **STRONGLY CONSIDER TREATMENT**
 **(STOP IF A NEGATIVE TEST)**

**Fig 2. A simple clinical risk score for empiric diagnosis of active TB.** Shown is a 1-page, easy-to-use tool for use in clinical settings where same-day microbiological testing for pulmonary TB is unavailable and risk of loss to follow-up is high. For illustrative purposes, we have chosen cutoffs of 10% and 40% risk of TB as potential clinical decision points, based on natural breaks in predictive probability and on intuition that TB treatment is unlikely to be started empirically for patients with less than a 1 in 10 chance of having TB. Other cutoffs could be selected based on local resources and probabilities of loss to follow-up if untreated. TB, tuberculosis.

The de novo model developed in the Ugandan population included a total of 9 points, of which 8 appeared in the primary model developed in the South African population (age, sex, HIV status, the number of WHO TB symptoms [1–4], and duration of symptoms) and 2 did not (antiretroviral therapy (ART) and cough observed during the interview). These 2 variables either were not collected (observed cough) or were rare (HIV not on ART) in the rural South African cohort. The c-statistic of this model in the internal validation population in Uganda was 0.81 (95% CI = 0.79 to 0.83) and was 0.78 (95% CI = 0.77 to 0.79) in the external validation population in South Africa (S5 and S6 Tables, S11 and S12 Figs).

After replicating non-cases in simulated cohorts, the sensitivity analyses showed no material change in discrimination; calibration was acceptable when the external validation population was modified in this fashion to match the prevalence of TB in the derivation population (see below), and estimates of clinical utility are provided across a range of underlying TB prevalence estimates to enhance transportability across settings. All estimates of variance, however, reflect only the primary data available (without replication of the Xpert–negative population).

## Discussion

This analysis of 2 different clinical populations in sub-Saharan Africa illustrates the successful development and validation of a clinical risk score for predicting the presence of active TB in

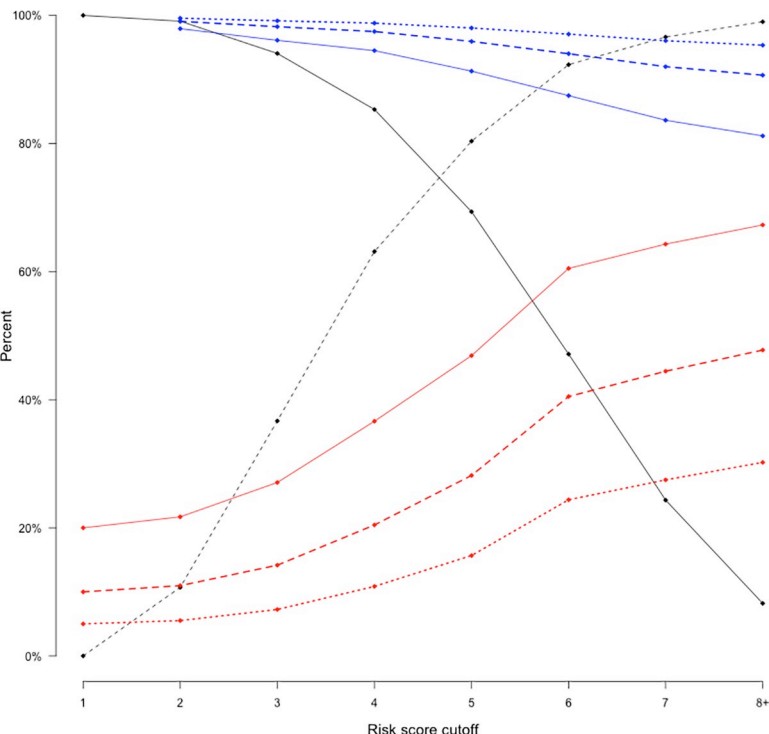

**Fig 3. Accuracy of a simple clinical risk score for active pulmonary TB.** This figure shows the sensitivity (solid black), specificity (dashed black), positive predictive values (red), and negative predictive values (blue) of a simple clinical risk score for pulmonary TB in the derivation South African cohort, at different cutoffs for a positive test. The positive and negative predictive values are estimated assuming a prevalence of TB among symptomatic individuals presenting for care (i.e., pretest probability) of 5% (dotted lines), 10% (dashed lines), and 20% (solid lines). The score ranges from a minimum of 1 to 10; see Fig 2. Due to the small sample size of individuals with scores higher than 8, we combined scores equal to or higher than 8 into 1 category. Accuracy and predictive values are calculated relative to Xpert MTB/RIF as a gold standard. TB, tuberculosis.

primary health clinics where same-day microbiological testing may be unavailable. This score is easy to calculate by hand and relies only on data that are readily available in the clinical setting. Despite its simplicity, the clinical risk score showed reasonable predictive accuracy, including in an external validation population. Use of this score was estimated to add clinical utility for patients in whom the benefits of an immediate diagnosis were similar or somewhat greater than the risks of false-positive diagnosis. De novo construction of a similar tool in the validation population yielded remarkably similar results, further suggesting that this clinical tool might be transportable across sub-Saharan African primary care settings. We have constructed a corresponding 1-page clinical tool that can be easily implemented in such settings. This clinical tool could reduce losses to follow-up (and corresponding transmission and mortality) by facilitating immediate initiation of treatment among patients who have a high clinical probability of having TB. Before recommending wide use, however, this tool must first be validated in other populations, and a strategy of using such a score to inform empiric treatment decisions must be evaluated in terms of its implementation and impact on patient outcomes (e.g., pretreatment loss to follow-up, treatment success, and mortality) in field settings.

A major potential benefit of this clinical TB score is its reliance on characteristics—namely younger age, male sex, number of TB symptoms (and duration >2 weeks), and HIV status—that are (1) easy to measure; (2) known to be associated with TB; and (3) confirmed to enhance predictive accuracy in 2 very different clinical populations. Our risk score formally combines

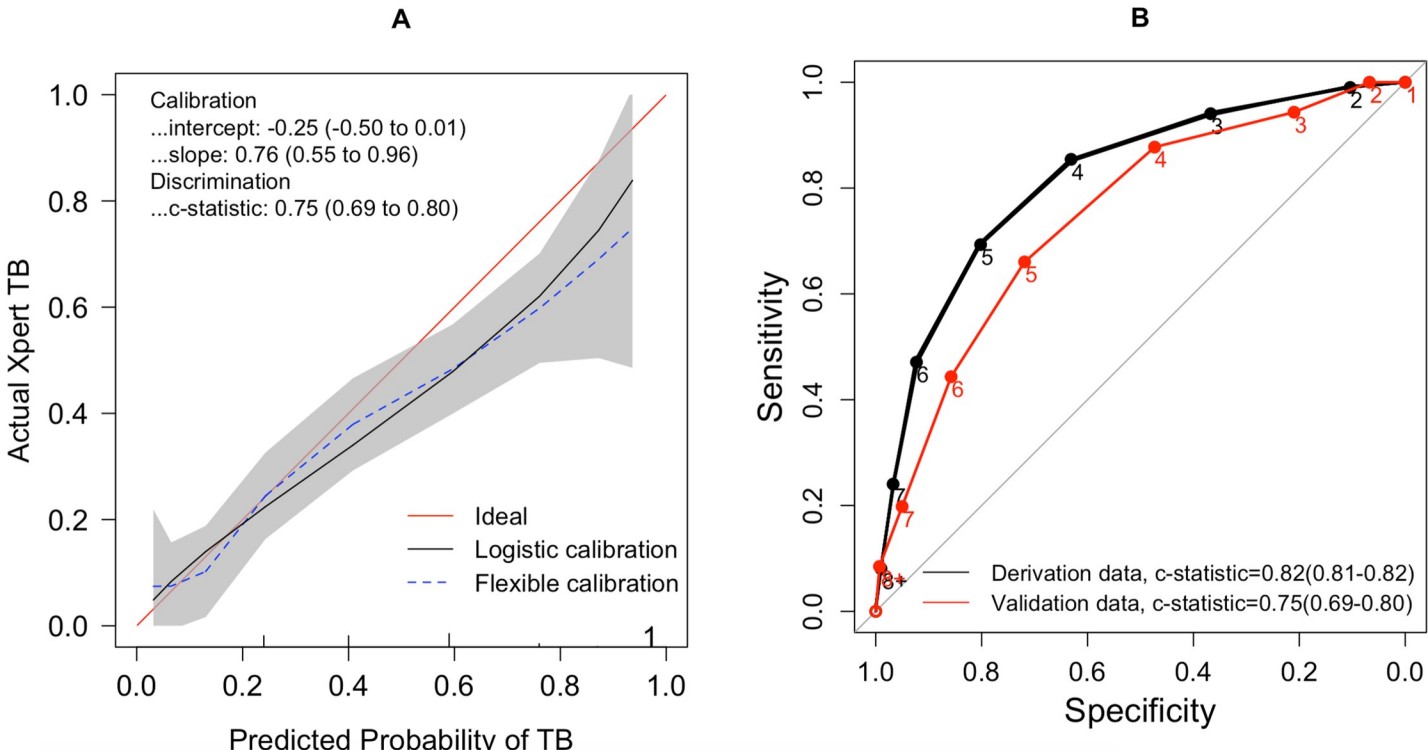

**Fig 4. Calibration and discrimination of a simple clinical score for diagnosis of active TB in sub-Saharan Africa.** Panel A shows model calibration using Cox linear logistic regression in the external validation (Ugandan) population. An intercept of 0 and slope of 1 is consistent with good calibration. In this plot, the red line represents perfect calibration, the black line corresponds to calibration of the simple clinical score, the dotted blue line corresponds to a smoothed (Loess) calibration, and the gray region corresponds to the 95% confidence band of the Loess calibration. The black line falling below the red line indicates that the simple score mildly overestimates the probability of TB in the validation population. Calibration curves were generated after adjusting for the different sampling fraction of TB in the derivation and validation populations, as described in the text. Panel B shows the ROC curve—a measure of discrimination—in the South African derivation (black line) and Ugandan external validation (red line) cohorts. Due to the small sample size of individuals with scores over 8, we combined scores equal to or higher than 8 into 1 category. The number on each dot represents the risk score at which sensitivity and specificity are estimated. For example, at a score of 4, specificity and sensitivity are 0.63 and 0.85, respectively, in the derivation population and 0.47 and 0.88, respectively, in the validation population. The reported c-statistics did not differ with adjustment of the sampling fractions to a population with 10% estimated prevalence. ROC, receiver operating characteristic; TB, tuberculosis.

these characteristics in a way that would be easy to calculate in the clinical setting and also easy to incorporate into policy. Regarding age, young adults may not experience higher absolute TB risk compared to older adults. Our data rather may reflect that alternative diagnoses (e.g., chronic obstructive pulmonary diseases and cancer) are less likely in younger patients who are seeking care for severe illness. Self-reported diabetes status contributed to the clinical score but may be slightly more difficult to standardize. Furthermore, the prevalence of diabetes and obesity is higher in South Africa than in other sub-Saharan African countries [30] and may not contribute substantially to predictive accuracy in those settings; as evidence of this, self-reported diabetes did not appear in the de novo model developed from the Ugandan population.

It is helpful that a simple clinical score performed nearly as well as a model using full regression coefficients, that imputation of the most common value (as might be done in the field) performed equally well as multiple imputation, and that predictive accuracy was not substantially degraded in external validation. Also reassuring is that the scores developed de novo in 2 separate populations converged on the majority of characteristics included. While these results imply the potential transportability of this clinical TB score to other populations (especially other populations seeking primary care for TB symptoms in sub-Saharan Africa), further

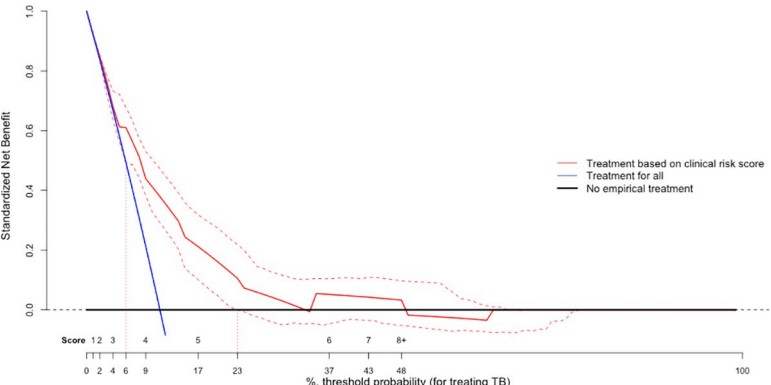

**Fig 5. Clinical utility of a simple clinical score for diagnosis of active TB in sub-Saharan Africa.** A decision curve analysis compares the standardized net benefit of different treatment strategies. The standardized net benefit was estimated as total benefit (treating true TB) minus total harm (treating false-positive TB), standardized to a maximum benefit of 1, assuming a population with 10% underlying prevalence of TB. The standardized net benefit was examined under different considerations of the relative benefit of a true-positive diagnosis versus the risk of a false-positive diagnosis (x-axis). This x-axis or threshold probability corresponds to the posttest probability given a TB prevalence of 10% among people being tested (i.e., as shown in Fig 2) at which the harms and benefits of empiric TB treatment are considered to be equivalent (i.e., the number of people without TB started on empiric treatment (false-positives) that would be tolerated to start empiric treatment on 1 additional person with TB (true-positive). The black numbers on top of the x-axis are the scores that correspond to each of the posttest probabilities. A treatment strategy with the highest net benefit at the particular threshold probability has the highest clinical value. The decision curve is based on the external validation population, with 95% confidence bands shown as dotted lines. Areas on the x-axis where the lower bound of the 95% CI is higher than the "treatment for all" line (i.e., threshold probability > 6%) offers a statistically significant benefit over treating all individuals. Areas on the x-axis where the lower bound of the 95% CI does not cross 0 (i.e., threshold probability ≤ 23%) illustrate settings in which use of the clinical risk score offers a statistically significant benefit relative to no empirical treatment. Numbers corresponding vertical dotted lines denote the threshold probabilities where the lower 95% confidence limit of "treatment based on clinical risk score" line is higher than the expected net benefit of treatment for all (blue line) or no empiric treatment (black line). These threshold probabilities (6%, 23%) include a threshold for treatment of a risk score of ≥4 (9% probability of TB) or ≥5 (17% probability of TB). Use of the clinical risk score would therefore offer higher net benefit than alternative treatment strategies (e.g., treatment for all or treatment for none) if the benefit of empirically treating someone with TB is deemed to be 3.3 to 15.7 times greater than the harm of empirically treating someone without TB. CI, confidence interval; TB, tuberculosis.

studies are required to confirm such generalizability before incorporating this score into widespread practice.

The value of this clinical score for active TB must be interpreted in light of its intended use. This score is only meant to guide management while awaiting microbiological (e.g., Xpert) results, not intended as a community-based triage test (i.e., to identify individuals who should receive further testing). Thus, while the score does not meet minimum WHO target product profile criteria for a community-based TB triage test (90% sensitivity and 70% specificity) [31], its value should be assessed not on its use as a triage test, but rather on the basis of its potential clinical utility in patients for whom a microbiological test is indicated but may not reliably or rapidly return. For example, in settings (e.g., high-income countries and most cities in middle-income countries) where microbiological results are likely to be consistently and rapidly available, a clinical prediction rule may not add substantively to the existing standard of care. If implemented in clinical settings, it is possible that this score could be misused (i.e., to forego microbiological testing in high-risk patients); any implementation plan would need to be accompanied by strong instructions and perhaps incentives to prevent such misuse.

However, in highly resource-constrained settings where Xpert results may not consistently return, this clinical score may help guide management. Many such settings employ midlevel clinicians and utilize highly standardized approaches to care. If a patient presents with TB

symptoms, for example, a sputum specimen will typically be collected, but treatment for TB will generally be deferred until the corresponding results return. In this context, pretreatment loss to follow-up is common [2]. If initiation of empirical TB treatment could reduce long waiting times, repeated visits, and delays in receiving results [2], it might facilitate immediate initiation of treatment (with the aim of reducing losses to follow-up) for high-risk patients. Indeed, many known risk factors exist for pretreatment losses to follow-up, including long turnaround time and travel times, high patient costs, and perceived stigma [2,32,33]. Clinicians might therefore selectively apply this score for patients with high risk of loss to follow-up. Alternatively, a similar algorithm to predict the risk of pretreatment loss to follow-up might be useful. Key priorities for future studies are therefore to better prospectively identify patients at high risk of pretreatment loss to follow-up and to better quantify the risk–benefit ratio of initiating empiric TB treatment, as functions of TB risk (as described here), clinical severity, and risk of loss to follow-up [34].

In selecting an appropriate cutoff for intervention on the basis of this clinical score, certain tradeoffs must be considered. If using a low cutoff of empiric treatment (for example, > score of 4 in a setting of ≥10% TB prevalence, yellow cells in Fig 2), the vast majority of individuals above this cutoff will not have TB. In such a setting, it would be critical to ensure that TB therapy could be de-escalated if microbiological results came back negative and clinical improvement were not consistent with TB (e.g., rapid improvement in a matter of days or no improvement over weeks). If such practice could be followed, same-day initiation of empiric TB treatment might improve clinical outcomes and reduce losses to follow-up, as seen with rapid initiation of ART for HIV [35,36]. Decision curve analysis (Fig 5) suggests that this is the context in which the score is likely to add greatest value, but future prospective studies would be necessary to confirm whether better outcomes could actually be achieved using this approach. If de-escalation of care is thought not to be feasible, an alternative use of this score would be to initiate empiric TB treatment only for those patients with a very high probability of TB (for example, 40% probability as defined by the red cells in Fig 2). In this context, future de-escalation of treatment might be less essential, but as suggested by the decision curve, this approach might not add substantial value over the current standard of care in which many such patients will already be treated.

The predictive accuracy of this simple clinical score was compatible with, or even superior to, previously published prediction models that incorporated criteria including chest X-ray [12] (c-statistic = 0.70, 0.79), body mass index (BMI) [13,15] (c-statistic = 0.70), hemoglobin level [14,15] (c-statistic = 0.66), HIV viral load and ART [37] (c-statistic = 0.59, 0.69), CD4+ T cell count [12,13,37] (c-statistic = 0.70), and Karnofsky score [14] (c-statistic = 0.75). These measures may not be readily available in highly resource-constrained settings that may lack even reliable scales and stadiometers. Of note, some prediction rules [12,37] have been explicitly designed for use in HIV–positive populations, in whom TB is often more difficult to diagnose. As such, the expected performance of these prediction rules might be lower, and they might be limited in their ability to generalize to HIV–negative populations. Some other prediction rules [38,39] have made in the context of contacts where not many of those experience TB symptoms. Our aim was to expand on this work by developing a tool that could be readily employed in peripheral health settings where a large fraction of individuals with TB symptoms may initially present to care [10], but where resources are extremely limited.

Our study has limitations. First, the South African derivation cohort included only 60% of all individuals who presented with TB symptoms during the study period. The majority of non-participants (70%) were not reachable by phone follow-up for collection of additional data. While the external validation population had much lower refusal rates (3%), these losses might induce bias in our results. Second, both the development and validation cohorts

included a random sample of people without TB, rather than evaluating a full cohort—and the sampling fractions differed between the derivation and validation cohorts. Non-cases were sampled to be representative of all people presenting with symptoms (i.e., not matched to cases), and we confirmed all results in simulated cohorts designed to match the underlying prevalence of TB; these analyses help to mitigate against spectrum bias but cannot remove it entirely. Third, we recruited people with presumptive pulmonary TB who presented to primary healthcare facilities. This clinical TB score may therefore not be applicable to such as active case finding where many people do not have TB symptoms, hospitals where individuals are acutely ill, or settings for predicting extrapulmonary TB. The utility of this score may also depend strongly on the underlying population's access to care—which can affect both symptoms on presentation and probability of returning for confirmatory microbiological test results [2]. Fourth, we omitted some factors that are known to be associated with increased TB risk (e.g., self-reported weight, clinician's subjective assessment of underweight status, or recent contact with a known TB case). These variables were either not measured in our population (self-reported weight and height and subjective assessment of underweight) or not found to improve predictive accuracy (recent TB contact). We did measure BMI in Uganda and found underweight status (BMI $< 18.5 \text{kg/m}^2$) as an additional score of 2, to improve discrimination; future studies should consider including these variables where height and weight can be reliably measured and BMI readily calculated. Fifth, the sensitivity of Xpert is imperfect, and many individuals in both cohorts were treated for TB without a positive Xpert result. By excluding these individuals from our analyses, our results may be biased in favor of better predictive accuracy (as these individuals may have higher clinical risk scores but may not have true underlying TB), may be subject to spectrum bias (as the clinical presentation of Xpert–positive and Xpert–negative TB cases may differ), and may exclude the very population for whom the clinical score might be most useful. This bias is likely to be small in the South African population because the prevalence of unknown Xpert status or empiric treatment was <1%. Sixth, this risk score may not be transportable to settings with aging populations, high prevalence of TB among non-native–born populations, or lower HIV prevalence. Finally, although the prediction model is simple, calculation of posttest probabilities requires knowledge of pretest probabilities (i.e., the proportion of patients tested with Xpert at a given facility who test positive); these numbers may not be widely known by treating clinicians.

In summary, we have developed a simple clinical TB score to inform empirical treatment decisions in resource-limited peripheral health settings of sub-Saharan Africa. This score is easy to calculate, consists only of easily measured variables, and performed well in 2 distinct populations (rural South Africa and urban Uganda). Use of this score may help to guide clinical management while awaiting results of microbiological testing, especially in highly resource-constrained settings where Xpert results may not be reliably available, and pretreatment losses to follow-up are high.

## Supporting information

**S1 Checklist. Transparent reporting of a multivariable prediction model for individual prognosis or diagnosis (TRIPOD) guideline.**
(DOCX)

**S1 Fig. Photos of the study sites.** Top is the study site in Limpopo Province, South Africa. Limpopo Province is widely rural and has a low population density (46 people per $\text{km}^2$). It has high TB incidence (301 per 100,000) and high adult HIV prevalence (17%). Two photos in the bottom are the study site in the capital city, Kampala, Uganda. In our Ugandan study site, the population density was estimated 23,000 people per $\text{km}^2$, TB prevalence was estimated 939 per

100,000 by GeneXpert Ultra catridge (418 per 100,000 after excluding GeneXpert Ultra trace positive but culture negative), and HIV prevalence was 18% among adults.
(TIFF)

**S2 Fig. A simple clinical risk score for empiric diagnosis of active tuberculosis in other pretest TB probability settings.** For illustrative purposes, we have chosen cutoffs of 10% and 40% risk of TB as potential clinical decision points, based on natural breaks in predictive probability and on intuition that TB treatment is unlikely to be started empirically for patients with less than a 1 in 10 chance of having TB. Other cutoffs could be selected based on local resources and probabilities of loss to follow-up if untreated. The utility of this score would markedly decrease in settings where the pretest TB probability is below 2.5% because the score higher than 7 is ≥15% at 2.5% TB setting.
(TIFF)

**S3 Fig. Calibration of a clinical risk score using actual regression coefficients rather than a simple 1 to 10 score.** The figure shows model calibration using Cox linear logistic regression in the external validation (Ugandan) population. An intercept of 0 and slope of 1 is consistent with good calibration. In this plot, the red line represents perfect calibration, the black line corresponds to calibration of the simple clinical score, the dotted blue line corresponds to a smoothed (Loess) calibration, and the gray region corresponds to the 95% confidence band. Calibration curves were generated after adjusting for the different sampling fraction of TB in the derivation and validation populations, as described in the main text.
(TIFF)

**S4 Fig. Discrimination of a more detailed clinical risk score using actual regression coefficients rather than a simple 1 to 10 score, including age modeled as a restricted cubic spline (RCS).** Panel A shows the receiver operating characteristic (ROC) curve—a measure of discrimination—in the South African derivation (black line) and Ugandan external validation (red line) cohorts. The reported c-statistics did not differ with adjustment of the sampling fractions to a population with 10% estimated prevalence. Panel B investigates the nonlinear relationship between the probability of Xpert MTB/RIF result and the continuous variable of age. The R package "Hmisc" was used to plot the estimated restricted cubic splines (RCS; with 5 knots placed at the natural quantiles of age). Arrows on the x-axis show the location of the knots. Of note, the second knot fell very near the median of our highest-risk category (i.e., age 25 to 44 years). Dots show a plot of nonparametric estimates (or smoothing parameters). The solid line shows the estimated spline transformation, and the dotted line shows the 95% confidence lines. Panel C compares the ROC curves using the actual regression coefficients; one with the age modeled as an RCS (solid line) and the other (dotted line) with the age modeled as categorical by decade (i.e., 15 to 24, 25 to 34, 35 to 44, 45 to 54, and 55+). The discrimination of the full model with age as a categorical variable was significantly higher than with age modeled as an RCS but was not statistically different from that of the simple 1 to 10 risk score presented in the main text.
(TIFF)

**S5 Fig. Discrimination of the model using each of actual classical TB symptoms as individual binary variables, instead of the total number of TB symptoms.** This receiver operating characteristic (ROC) curve shows the discrimination of the model using each of actual classical TB symptoms as individual binary variables, instead of the total number of TB symptoms. The black curves show the clinical score's discrimination in the derivation (South African) population, and the red curves show the corresponding analysis in the external validation (Ugandan) population. The number on each dot represents the risk score at which sensitivity and

specificity are estimated. The discrimination of the model using each symptom individually was not significantly higher than that of the main simple risk score in which symptoms were simply counted, shown in Fig 4B.
(TIFF)

**S6 Fig. Discrimination of the model within individuals who presented chronic cough regardless of any other TB symptoms.** In South African population, 665 (49%) reported chronic (>2weeks) cough, 61% of which were Xpert–positive TB cases in South African population. In Ugandan population, 262 (68%) reported chronic cough in Ugandan population, 36% of which were Xpert–positive TB cases. This receiver operating characteristic (ROC) curve shows the discrimination of the model within individuals who presented chronic cough regardless of any other TB symptoms. The black curves show the clinical score's discrimination in the derivation (South African) population, and the red curves show the corresponding analysis in the external validation (Ugandan) population.
(TIFF)

**S7 Fig. Calibration prior to adjustment for different sampling fractions.** The clinical risk score for active tuberculosis (TB) as derived in the South African population is expected to systematically overestimate the predicted risk of TB in the external validation Ugandan population because of the different sampling fractions used. (Specifically, in South Africa, approximately 1 Xpert-individual was sampled per case, whereas in Uganda, nearly 2.7 Xpert–negative individuals were sampled per case.) To appropriately evaluate model calibration, therefore, we adjusted the intercept of the Cox linear–logistic model by multiplying the log odds ratio of TB in the validation population versus the derivation population. This intercept is a measure of the difference between the predicted probability and the actual outcome (i.e., calibration-in-the-large). Adjusting the intercept (for everyone in the study population) does not affect the order of predicted risks of individuals, and hence, it does not change the predictive accuracy in terms of discrimination (i.e., c-statistics). Panel A shows model calibration using Cox linear logistic regression in the external validation (Ugandan) population before adjusting the intercept, using simple scores as the prediction model. An intercept of 0 and slope of 1 is consistent with good calibration. In this plot, the red line represents perfect calibration, the black line corresponds to calibration of the simple clinical score, the dotted blue line corresponds to a smoothed (Loess) calibration, and the gray region corresponds to the 95% confidence band. The divergence between the black and red lines illustrates the poor calibration prior to adjustment of the intercept; as described above, this poor calibration is expected on the basis of different sampling fractions. Panel A shows calibration of the simple 1 to 10 score, whereas Panel B shows calibration of the model using full regression coefficients. Both of these plots were generated in the external validation (Ugandan) population before adjusting the intercept.
(TIFF)

**S8 Fig. Discrimination of the clinical risk score with missing values imputed as the most common value rather than by multiple imputation.** This receiver operating characteristic (ROC) curve shows the discrimination of the model after imputing missing data with the most common value (as might be done in clinical practice), rather than using multiple imputation. The black curves show the clinical score's discrimination in the derivation (South African) population, and the red curves show the corresponding analysis in the external validation (Ugandan) population. The solid lines and dotted lines show the simple 1-to-10 scoring and scoring system using full regression coefficients, respectively. Comparing this figure to Fig 4B in the main text illustrates that the simple risk score retains the same discrimination (c-statistic

0.75) in the external validation population when using a simple imputation technique (replacing missing values with the most common value in the population), which could be performed easily in clinical practice.
(TIFF)

**S9 Fig. Discrimination of the clinical risk score after including empirically diagnosed TB patients as TB positive or as TB negative.** In Ugandan study population, 27 TB cases were clinically diagnosed; 16 of which were empirically treated without microbiological confirmation and 11 of which were empirically treated regardless of negative Xpert results. Among 27 patients treated empirically, 7 (26%) patients ultimately had Xpert–positive (4/7) or culture positive (3/7). The risk score of empirically diagnosed individuals (median = 6, IQR = 5–6) was as high as the score of Xpert-confirmed TB cases (5, 4–7) compared to the Xpert–negative group (4, 3–5). The receiver operating characteristic (ROC) curves show the discrimination of the model after including empirically diagnosed TB patients in the study population. Left panel is the ROC curve when empirically diagnosed TB patients were considered as TB positive together with Xpert-confirmed TB-positive cases. Right panel is the ROC curve when empirically diagnosed TB patients were considered as TB negative. The curve shows the clinical score's discrimination in the external validation (Ugandan) population. The points on the line show the simple 1-to-10 scoring. Comparing these figures to Fig 4B in the main text illustrates that the simple risk score retains the similar discrimination power (c-statistic 0.76 versus 0.75) when treating empirically diagnosed TB patients same as Xpert-confirmed TB-positive cases. As expected, including these empirically diagnosed individuals (slightly) increased the predictive power of our risk score.
(TIFF)

**S10 Fig. Clinically utility of a simple clinical score for diagnosis of active tuberculosis in sub-Saharan Africa assuming a population with 5% underlying prevalence of TB.** The standardized net benefit (y-axis) was estimated as total benefit (treating true TB) minus total risk (treating false-positive TB), standardized to a maximum benefit of 1, assuming a population with 5% underlying prevalence of TB. The standardized net benefit was examined under different considerations of the relative benefit of a true-positive diagnosis versus the risk of a false-positive diagnosis, or threshold probabilities (x-axis). The decision curve is based on the external validation population, with 95% confidence bands shown as dotted lines. Black numbers on top of the x-axis are the posttest probability we estimated in the main Fig 1 under 5% pretest probability. Red numbers correspond to the threshold probabilities where the lower 95% confidence limit of "treatment based on clinical risk score" line is higher than the upper 95% confidence limit of other lines of 2 different treatment strategies. Use of the clinical risk score of 4 or 5 would offer higher net benefit than alternative treatment strategies (e.g., treatment for all or no empiric treatment) between 3% and 11%. This figure can also be used to evaluate the potential impact of incorrectly assumed pretest probability of TB—for example, if practitioners believe that the pretest probability is 10%, when in reality it is 5%. In this example, the only score in which a misspecification would lead to an incorrect treatment decision is a score of 5. This implies a posttest probability of 17% and an empiric treatment consideration at a 10% pretest probability setting, when in fact the posttest probability is 9% and no empiric treatment consideration required at a 5% pretest probability setting. About 20% of symptomatic individuals may be falsely treated until the microbiological testing confirmation is available.
(TIFF)

**S11 Fig. Calibration of a simple clinical score for empirical diagnosis of active tuberculosis developed from the Ugandan study population.** Panel A shows model calibration using Cox

linear logistic regression in the internal validation (Ugandan) population, among those whose data did not contribute to model development. An intercept of 0 and slope of 1 is consistent with good calibration. In this plot, the red line represents perfect calibration, the black line corresponds to calibration of the simple clinical score, the dotted blue line corresponds to a smoothed (Loess) calibration, and the gray region corresponds to the 95% confidence band. Panel B shows model calibration in the external validation (South African) population. The calibration curve for Panel B was generated after adjusting for the different sampling fraction of TB in the derivation and validation populations, as described in the text.
(TIFF)

**S12 Fig. Discrimination of a simple clinical score and more detailed clinical risk score (using actual regression coefficients) for empirical diagnosis of active tuberculosis developed from the Ugandan study population.** Panel A shows the receiver operating characteristic (ROC) curve in the South African derivation cohort (red line), internal validation cohort (blue line), and Ugandan external validation cohort (black line). The number on each dot represents the risk score at which sensitivity and specificity are estimated. For example, at a score of 5, sensitivity and specificity are 0.66 and 0.82, respectively, in the derivation and internal validation populations, versus 0.79 and 0.64, respectively, in the external validation population. The reported c-statistics did not differ with adjustment of the sampling fractions to a population with 10% estimated prevalence. Panel B shows the ROC curve using more detailed score incorporating full regression coefficients. The discrimination power was retested under an assumption of 10% TB prevalence (by replicating the TB-negative population to fit the target prevalence); the c-statistics remained the same.
(TIFF)

**S1 Table. Interview questions for self-reported tuberculosis symptoms and HIV status.**
(DOCX)

**S2 Table. Observed and predicted probability of active pulmonary TB in the Ugandan external validation population.** The table provides the observed percentage of the population in the external validation population (urban Uganda) who had Xpert-confirmed TB, given each clinical risk score. The observed percentages are compared to the percentages predicted by the clinical risk score (right-most column). The optimal cutoff for clinical decision-making was at a score of $\geq 4$ or $\geq 5$; above these cutoffs, observed and predicted probabilities were similar.
(DOCX)

**S3 Table. Lasso regression coefficients and the simple scoring system after modeling actual classical TB symptoms as individual binary variables (instead of the total number of TB symptoms).** We explored use of the actual regression coefficients from the lasso model (rather than a simple 1-to-10 scoring system), including both a categorical expression of age and a representation of age using restricted cubic splines. For each of these alternative scoring systems, derivation and external validation were performed in the same fashion as for the primary score described in the main text.
(DOCX)

**S4 Table. Lasso regression coefficient and simple scoring systems, using a single imputation method (i.e., mode) for the missing predictors.** For the clinical risk score to be useful as a tool in the field, clinicians must be able to directly impute missing data when those data are unavailable. We therefore performed a secondary analysis in which all missing values were assigned the most common value (or single imputation) to reflect likely clinical use of the

score in the field. For example, when a clinician does not have information about duration of TB symptoms, the clinician can use the dominant value, 0 or 1, among patients. In our study, a missing duration of TB symptoms, any other non-TB symptoms, and diabetes mellitus was replaced to 1, 1, and 0. After replacing the missing values with the most common value of each variable, the simple score rule based on the least absolute shrinkage and selection operator (lasso) regression coefficients remained the same. We then compared the discrimination of this model using simple (clinical) imputation versus multiple imputation.
(DOCX)

**S5 Table. Additional characteristics of the model derivation population (Uganda) after bootstrapping and random splitting of the model derivation and the internal validation population.** To further assess the transportability of the components of the clinical risk score, we performed a de novo model development using the population from Kampala, Uganda, as the derivation population and the population from South Africa as the external validation population. For this analysis, we used a 20-fold bootstrap sample of the Ugandan population (given its smaller sample size) to optimize statistical power. Then, we applied a split internal validation approach by randomly selecting two-thirds of the population as the model training cohort and internally validating the model on the remaining one-third of the testing population (whose data did not contribute to model development).
(DOCX)

**S6 Table. Association of key variables with Xpert-confirmed pulmonary tuberculosis in the bootstrapped Ugandan population.**
(DOCX)

## Acknowledgments

We would like to thank the patients for participating in the study and the field staff for supervising and coordinating data collection.

## Author Contributions

**Conceptualization:** Neil Martinson, Achilles Katamba, David W. Dowdy.

**Data curation:** Yeonsoo Baik, Hannah M. Rickman, Lesego Mmolawa, Peter J. Kitonsa, Tsundzukana Sewelana, Annet Nalutaaya, Emily A. Kendall, Limakatso Lebina.

**Formal analysis:** Yeonsoo Baik, Hannah M. Rickman.

**Funding acquisition:** Colleen F. Hanrahan, David W. Dowdy.

**Investigation:** Colleen F. Hanrahan, Emily A. Kendall, Neil Martinson, Achilles Katamba, David W. Dowdy.

**Methodology:** Yeonsoo Baik, Colleen F. Hanrahan, Emily A. Kendall, David W. Dowdy.

**Project administration:** Colleen F. Hanrahan, Lesego Mmolawa, Peter J. Kitonsa, Tsundzukana Sewelana, Annet Nalutaaya, Emily A. Kendall, Limakatso Lebina, David W. Dowdy.

**Resources:** Lesego Mmolawa, Peter J. Kitonsa, Tsundzukana Sewelana, Annet Nalutaaya, Emily A. Kendall, Limakatso Lebina, Neil Martinson, Achilles Katamba, David W. Dowdy.

**Supervision:** Colleen F. Hanrahan, Achilles Katamba, David W. Dowdy.

**Validation:** Lesego Mmolawa, Peter J. Kitonsa, Tsundzukana Sewelana, Annet Nalutaaya, Emily A. Kendall, Limakatso Lebina, Neil Martinson.

**Visualization:** Yeonsoo Baik.

**Writing – original draft:** Yeonsoo Baik, Hannah M. Rickman, David W. Dowdy.

**Writing – review & editing:** Yeonsoo Baik, Emily A. Kendall, Neil Martinson, Achilles Katamba, David W. Dowdy.

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
