## [Editor Report · Decision Letter 0]

19 May 2020

Dear Dr Baik, 

Thank you for submitting your manuscript entitled "A simple clinical score for predicting active tuberculosis when same-day microbiological testing is unavailable" for consideration by PLOS Medicine.

Your manuscript has now been evaluated by the PLOS Medicine editorial staff and I am writing to let you know that we would like to send your submission out for external peer review.

Kind regards,

Thomas J McBride, PhD,

PLOS Medicine

---

## [Decision Letter · Decision Letter 1]

22 Jun 2020

Dear Dr. Baik,

Thank you very much for submitting your manuscript "A simple clinical score for predicting active tuberculosis when same-day microbiological testing is unavailable" (PMEDICINE-D-20-01816R1) for consideration at PLOS Medicine. 

[LINK]

In light of these reviews, I am afraid that we will not be able to accept the manuscript for publication in the journal in its current form, but we would like to consider a revised version that addresses the reviewers' and editors' comments. Obviously we cannot make any decision about publication until we have seen the revised manuscript and your response, and we plan to seek re-review by one or more of the reviewers. 

We expect to receive your revised manuscript by Jul 13 2020 11:59PM. Please email us (plosmedicine@plos.org) if you have any questions or concerns.

We look forward to receiving your revised manuscript. 

Sincerely,

Emma Veitch, PhD

PLOS Medicine

On behalf of Clare Stone, PhD, Acting Chief Editor,

PLOS Medicine

plosmedicine.org

*We'd suggest revising the title per PLOS Medicine's style, ideally this should include the study design in the subtitle (after a colon), eg ": A randomized controlled trial," "A retrospective study," "A modelling study," etc.

*Please structure the abstract using the PLOS Medicine headings (Background, Methods and Findings, Conclusions; "Methods and Findings" should be a single subsection. In the last sentence of the Abstract Methods and Findings section, please include a brief summary of any key limitation(s) of the study's methodology.

*At this stage, we ask that you include a short, non-technical Author Summary of your research to make findings accessible to a wide audience that includes both scientists and non-scientists. The Author Summary should immediately follow the Abstract in your revised manuscript. This text is subject to editorial change and should be distinct from the scientific abstract. Please see our author guidelines for more information: https://journals.plos.org/plosmedicine/s/revising-your-manuscript#loc-author-summary

*Please clarify in the paper (Methods section) whether this analysis, reported in the paper, corresponded to that laid out in a prospective protocol or analysis plan? Please state this (either way) early in the Methods section.

*The authors may wish to consider using a reporting guideline corresponding to their specific study design to aid fuller reporting of the methods and findings of this paper. The TRIPOD statement may be the most appropriate (https://www.tripod-statement.org/TRIPOD/TRIPOD-Checklists) and the authors may like to look at whether this guideline is helpful; if so the checklist provided on the TRIPOD website could be used to assist reporting - in which case please fill out the checklist and upload as supporting information when resubmitting. In addition to this the authors might like to cite either the TRIPOD statement or other reporting guideline they used in enhancing study reporting. 

Comments from the reviewers:

Reviewer #1: Dear Authors,

Thank you for the opportunity to review this great and important paper, which I have recommended for publication. I attach a letter with some minor comments for consideration prior to publication but have let the editor know that I do not need to see another version of this manuscript.

Congratulations on your excellent, practical, research.

Yours sincerely,

Tom Wingfield

Reviewer #2: Comments to the authors

I enjoyed reading this manuscript, which makes an important and novel contribution to the literature. The study aims to address the critical question of how to manage patients presenting to primary care settings with TB symptoms for whom same day microbiological testing is unavailable. This is a thorough and impressive piece of work, with sound statistical methods that are strengthened significantly by the extensive additional analyses presented in the supporting information. Some of the issues I would have highlighted have already been addressed by the authors in these supplementary analyses or have been thoughtfully discussed in the limitations. However, I have a number of comments and suggestions that may help to further strengthen the manuscript that mostly relate to the clinical utility of the score. 

Cut offs and implications for practice. The authors present standard parameters used to evaluate diagnostic tests. However, I think that what the use of this score in practice might explicitly mean should be more clearly demonstrated, and the implications of specific strategies further considered. There is a slight contradiction in the manuscript because in the risk score figure, the authors are careful to frame clinical recommendations as "consider or strongly consider treatment". However, in the discussion, the authors highlight that in many resource-constrained settings mid-level clinicians utilize standardised approaches to care. If a standardised approach were to be used, using a score cut off of four for giving empirical TB treatment gives a sensitivity of 85% and a specificity of 63% (optimistic scenario using the derivation data). Translating these figures into a population of 1000 people where prevalence of TB in tested individuals is 10% means that 85 would be appropriately treated for TB, 15 would not be treated immediately despite having TB, and a further 333 would also be given TB treatment who subsequently have a negative Xpert result. Using a different cut-off point would mean different results and this trade-off between sensitivity and specificity is very important and could be further elaborated upon. If the score were to be applied in this standardised way with a cut off of four, this would mean a very large number of people starting TB treatment empirically who subsequently have a negative Xpert result (four times the number starting TB treatment who subsequently have a positive Xpert result). Some of these "negative but treated" people will truly actually have TB, as a single Xpert test on sputum has imperfect sensitivity for pulmonary TB and is inadequate to diagnose extra-pulmonary TB. However, in most cases people who tested negative for TB will truly not have TB and may have an alternative diagnosis such as a simple upper respiratory tract infection. Although the vision is that clinicians will be able to de-escalate treatment, I'm not sure this is likely to happen in practice and feel that once a decision to start TB treatment has been made, it is infrequent for that decision to be reversed, especially in the absence of microbiological/radiological testing. Furthermore, by the time clinicians evaluate people with a negative result to decide about TB treatment de-escalation, a simple upper respiratory tract infection might have spontaneously resolved, appearing to mimic a good response to TB treatment and increasing the chances of the clinician simply continuing TB treatment, inappropriately. An alternative approach of only treating people empirically who have very high-risk scores has its own limitations and I think that these complex trade-off issues merit further discussion in the manuscript. 

Loss to follow-up. As the authors state, the score is not designed and should not be used to replace definitive microbiological testing, but rather aims to identify people who are at high risk of TB to facilitate empirical management with the aim of reducing loss to follow-up for the highest-risk patients. Therefore, I think the utility of this score depends both on the risk of TB, and on the risk of loss to follow-up for that individual person. Perhaps the authors could discuss in further detail some of the reasons why people are lost to follow-up (e.g. long travel times, costs, stigma) and the potential for derivation of similar algorithms to predict loss to follow-up that could be used alongside their tool to inform optimal decision making. They might also reflect upon other strategies that could be used to reduce pre-treatment loss to follow-up such as incentives and social support interventions. Overall, I think that there is a clear need for more research to address these important questions because the decision to start empirical treatment versus waiting for results/illness evolution depends on many factors, including local epidemiology, severity of illness, availability of further testing/chest radiography, and risk of loss to follow-up.

Decision curve analysis. The use of a decision curve analysis is appropriate and interesting. However, I think that people inherently find the interpretation of these analyses and graphics confusing. I think the authors have done a good job in trying to explain and provide context to the findings, but I wonder if they might be able to go even further to make these findings more accessible to people not familiar with the analysis strategy. I also wonder whether "cost:benefit" should be rephrased as "risk:benefit". Obviously a false-positive diagnosis incurs economic costs to both the person and health system, but also more broader risks associated with inappropriate treatment, adverse drug reactions, and failure to diagnose alternative pathologies. As highlighted above, perhaps some of these trade-off issues could be further explored in the discussion. I was also unclear what statistically significant means in the context of the decision curve analysis, and whether 95% confidence intervals are necessary. 

Minor comments.

1. Apologies if I have missed it but it wasn't clear to me what symptoms triggered TB testing in the derivation cohort, and how this differed from how presumptive TB was defined to trigger TB testing in the validation cohort? Table 1 seems to suggest that people had to have at least one of cough, fever, weight loss, or night sweats to receive an Xpert test? Were the same criteria used in both cohorts? In some settings where this score might be most valuable, patients only receive a TB diagnostic test if they have productive cough for over two weeks. Could the authors potentially comment upon how this might affect the interpretation and use of their score?

2. Please provide more detail on the symptoms included in the model. Should clinicians ask about any cough of any duration, or specifically productive cough? Does weight loss have to be over a specified amount or is it simply self-reported? Did the authors have data available on haemoptysis? Do symptoms refer to a certain time period, i.e. if you have had one episode of fever three weeks ago but none since, does this count as having presented with fever?

3. Relatedly, how variables are measured is likely to impact on predictive performance. Diabetes and HIV are self-reported, but these diseases are substantially underdiagnosed in many of the settings where the authors propose the score could be used. I note that HIV status is known for all participants in this study, was that because rapid testing was done at recruitment and so clinicians had up to date results at the time of risk score assessment? What should be done in settings where somebody last had an HIV test three years ago and the assessing clinician does not have access to rapid testing now? Do we assume they are negative, potentially misclassifying them and underestimating their risk of TB? This is complex as in some sub-Saharan African countries HIV prevalence is extremely high, especially in people presenting to healthcare with symptoms. Similarly for diabetes. 

4. I think Table 1 should be checked and include missing data for each variable. Some of the numbers don't add up, e.g. duration of TB symptoms 206+465=671. But n=707 for the column. 

5. I think the number 1634 (total population in derivation cohort) is an error in the results. Also, the authors first report non-participation in the discussion. I think a flow chart demonstrating study design and recruitment would significantly add to the manuscript and help readers to understand what the authors did. Relatedly I would be interested to know what the prevalence of Xpert confirmed TB was in people presenting with symptoms at these clinics, to have a clearer understanding of what sampling fraction the authors recruited. How were the Xpert negative people randomly selected? These data and questions could all be easily presented and addressed in a figure.

6. I think a great strength of this study is the derivation and validation of the score in two diverse cohorts. Perhaps the authors could supplement the above suggested figure with some population-level data demonstrating the key differences between these two populations e.g. underlying HIV prevalence, poverty estimates. Photos may also be a nice addition.

7. I was surprised to see that both smoking and previous TB were associated with TB diagnosis in univariable analysis (as you would expect), but not in multivariable analysis. Could the authors comment upon which factors included in the final model are felt to be confounding or mediating these associations?

8. I agree with the importance of deriving a model that could be easily used in a resource-constrained primary health care clinic. However, I can think of a number of variables that might predict TB that have not been included, which would be easily measurable. For example, epidemiological factors such as if the person was known to have a recent TB contact. Relatedly, I think that although BMI might be difficult to measure in many settings, clinicians can easily visually estimate whether a person is underweight versus normal weight versus overweight. If BMI data are available to analyse then I would be interested to see if a variable of weight category predicts TB in these populations, and importantly whether it adds any predictive power to their model.

9. I think the discrimination of the score in the derivation cohort should be mentioned in the main text of the results and not just the figure. 

10. Are the sensitivity and specificity in figure 4 based on the performance of the score in the derivation cohort, or the validation cohort? I suspect derivation cohort from studying the data.

11. I think the last sentence of the first paragraph of the discussion should be reworded, emphasising the need for further operational/impact research to investigate the use of a score-based strategy. Further validation alone will not demonstrate that the score can reduce loss to follow-up, transmission, and mortality.

12. I would be interested to know if smear microscopy was available in these clinics and if the authors have data on this? This is still the most commonly used method to diagnose TB globally and is available in many of the settings where this score might be most useful. 

13. I think the sentence "These sensitivity analyses showed no material change in discrimination…." would be more appropriate in the results rather than the methods.

None of these comments should take away from the excellent and comprehensive work the authors have done. Please accept my apologies for the rather long review. 

Open review

Signed: Matthew Saunders

Reviewer #3: This paper addresses the important question of whether a simple clinical risk score can be used to guide empirical TB treatment in patients presenting to primary care in low resource settings, whilst waiting for the results of microbiological TB tests. The authors develop a pragmatic score using data from a cohort in rural South Africa, which is they then validate in a prospective study in a very different population in Uganda, and find it performs similarly. Overall the study is well designed, and the paper very well written.

Comments:

-It would be good if the authors expanded on how they randomly selected the Xpert-negative patients in both cohorts (this can be added to Supplement if needed). 

-The authors modified the intercept of the fitted regression model in the Ugandan cohort to mitigate for the different sampling fractions from the 2 cohorts. Is there a reason the sample fraction couldn't be increased from the SA cohort to be closer to the Ugandan cohort?

-Please can you further explain the Cox's linear-logistic regression model used in the validation cohort to assess calibration (it is not a regression model I am familiar with)

-Could the authors explain the decision curve analysis in more detail (again, this could go in the supplement if needed)? Specifically, a brief explanation of threshold probability and benefit may help the readers

-Linked to this, in figure 2 I suggest the authors present threshold probability as a %. Can the authors also clarify of the 'treatment for none' line is mean to be 'no empirical treatment' as is suggested in the figure legend (his maybe misinterpreted to mean no TB treatment, even if subsequent positive TB tests). it may also be clearer to colour the areas relating to the 'benefit over treating all' and 'benefit to no treatment'.

-Can the author's justify including a 20% TB prevalence in the models- it seems too high to me for TB prevalence in primary care settings (even amongst patients with TB symptoms). Is the TB prevalence in the 2 cohorts known? If not please cite relevant recent studies to justify TB prevalence in primary care settings. It may be make more sense to include a lower pre-test probability in the clinical tool and analyses. 

-Do the authors have data on how many patients in the 2 cohorts were started on TB treatment 'empirically' (ie prior to microbiological test result)? It would be useful to compare the performance of the score to clinicians decision making on empirical treatment without a score.

-Similarly, the authors recognise the exclusion of patients treated for TB in the absence of positive microbiological results as a limitation. Can the authors report how many patients in each cohort were treated for TB but had negative Xpert and/or other microbiological TB tests? How many patients had radiological evidence of TB but negative tests? IT may be worth considering including these patients in another sensitivity analysis.

-There is a significant difference between self-reported diabetes prevalence in the South African and Ugandan cohorts (I suspect this will be similar for many settings outside South Africa, or in more rural settings). Can the authors expand on this, including reasons why it might be the case, in the discussion

-The authors used a simple imputation model for missing data using the mode value. Can the authors expand on how this translates to use in the field- eg if HIV status is unknown, how should that be scored when calculating risk score (maybe this could be added to figure 3 as a footnote)?

-Figure 4 would benefit from a legend explaining the different lines

Reviewer #4: "A simple clinical score for predicting active tuberculosis when same-day microbiological testing is unavailable" describes the derivation of a simple clinical score for use in assessing risk of tuberculosis (TB). This clinical score is expressedly designed not to require any computational resources, so as to be rapidly tallied by hand if needed, using just six broad [categories of] features (Table 2; age, sex, HIV status, classical TB symptoms, symptom duration, diabetes). Therefore, it is particularly suitable for deployment in under-resourced countries.

Initial model derivation was done using training data from 28 clinics in rural South Africa. Regression was performed on training data, and regression coefficients were converted into a scoring scheme, with further calibration and checks of clinical utility performed. External validation was performed on four clinics in urban Uganda. Simulation of different TB prevalence levels was performed as sensitivity analyses on the robustness of the model during validation, and comprehensive de novo analyses were performed to assess generalizability of the model construction (Appendix Section V)

It may be noted that a fair number of similarly-targeted low-resource clinical scores for TB prediction have been proposed previously, as discussed in the paragraph from Line 350 onwards. This proposal claims to be an improvement by not requiring criteria that require additional equipment as far as possible (e.g. scales, for body mass index), and also to provide rules that are also more generally applicable, rather than particularly to HIV-positive populations.

1. The authors might consider emphasizing that this clinical score is most appropriate for a sub-Saharan African population in the manuscript title (since this is where the training/validation data was exclusively drawn from), unless it can be convincingly shown to apply generally.

2. The authors might consider introducing and emphasizing the actual proposed scoring system (Figure 3) early on in the text, since it would appear to be of overriding interest for interested readers.

3. A brief description of the expected accuracy/sensitivity/specificity of the Xpert sputum test considered as gold standard here, might be included in the methods (it is stated in the limitations that Xpert is an imperfect test)

4. Appendix Section IV describes single imputation (with the most common value for each predictor), as likely practice in the field. However, it is not clear as to what the modal value for each predictor is, either in Figure 3 or the relevant tables. These model "default" values might be included (and explained) in Table 2/Figure 3.

5. The definitions of the classical WHO TB symptoms (cough/fever/night sweats/weight loss) appear to be rather underspecified. Is there an unspoken understanding amongst clinicians as to what qualifies for each of these symptoms, or are these symptoms expected to be wholly based on self-reporting? For example, the participant questionnaire for the Kharitode TB study [16] data that this publication is based on appears to specifically define weight loss as "more than 5 kg, or enough to make my clothes loose" - might this be relevant here?

6. Likewise, the clarity of the "Symptom duration > 14 days" predictor might be made more explicit, as to whether it refers to any symptom (assumed), or to some/all (i.e. Question 15 or 16 of Kharitode TB questionnaire)?

7. For Table 2, while the intent behind the score derivation is clear, the details remain slightly opaque (i.e. footnote c: "Each point in this simple clinical score is estimated by dividing the corresponding LASSO coefficient by the median value of four clustered coefficients (0.9) and rounding to the nearest integer..."); this might be explained slightly further - in particular, what is the significance of the four clustered coefficients, and of their median?

8. Figure 2 on cost-benefit decision curve analysis appears to indicate that the proposed clinical scoring system has potential to provide additional utility over basic "treat all" and "treat none" strategies (it is assumed that these costs & benefits are defined in economic terms?). For completeness, the relevant assumed values for costs & benefits given each patient scenario (i.e. "treat" and "TB", "treat" but "no TB", "no treat" and "no TB", "no treat" but "TB") might be specified and justified.

9. On Figure 2, it appears that various points on the "treatment based on clinical risk score" curve correspond to discrete patient scores; if so, the authors might consider labelling these scores on the curve, as with Figure 1 Panel B.

10. The reported predictive accuracies of the various previous models mentioned from Line 350 onwards, might be provided in the text for convenience.

11. The limitation that the proposed clinical scoring system should not be intended for community-based triage (Line 328) appears fairly significant, and might be emphasized in the abstract.

12. The limitation on requiring pre-test probability of true/Xpert TB appears fairly major in practice. The authors might consider further motivating the scoring system for decision curve analysis based on different assumed pre-test TB probabilities (i.e. Figure 2, for 5%/20% TB probability), and with incorrectly-assumed TB probabilities (i.e. assumed 10% probability, actually 20%). This is because it appears likely that the pre-test probability might not be estimated particularly accurately, especially where clinics do not encounter large numbers of patients.

Minor comments:

13. In Line 356, it is stated that "As such, the expected performance of these prediction rules might be lower, but they might also be limited in their ability to generalize to HIV-negative populations"; the authors might consider clarifying that the lower performance is due to the HIV-positive population, and/or change the "but" to "and".

[LINK]

---

## [Decision Letter · Decision Letter 2]

23 Sep 2020

Dear Dr. Baik,

Thank you very much for re-submitting your manuscript "A simple clinical score for predicting active tuberculosis when same-day microbiological testing is unavailable: a multivariable prediction model using data from a clinical trial in South Africa and a population-based survey in Uganda" (PMEDICINE-D-20-01816R2) for review by PLOS Medicine.

I have discussed the paper with my colleagues and the academic editor and it was also seen again by three of the original reviewers. I am pleased to say that provided the remaining editorial and production issues are dealt with we are planning to accept the paper for publication in the journal.

[LINK]

We look forward to receiving the revised manuscript by Sep 30 2020 11:59PM. 

Sincerely,

Thomas McBride, PhD

Senior Editor 

PLOS Medicine

plosmedicine.org

Requests from Editors:

1- Thank you for editing the title, please revise further to: “A clinical score for identifying active tuberculosis without microbiological testing: development and validation of a multivariable prediction model”

2- Second point of Author Summary, “*require* data…”

3- Thank you for including the TRIPOD checklist. Please add the following statement, or similar, near the start of the Methods: "This study is reported as per the Transparent reporting of a multivariable prediction model for individual prognosis or diagnosis (TRIPOD) guideline developing, validating, or updating a prediction model (S1 Checklist). Please also replace the page numbers in the checklist with paragraph numbers per section (e.g. "Methods, paragraph 1"), since the page numbers of the final published paper may be different from the page numbers in the current manuscript.

4- Thank you for noting your data are available. At this time, please update your data statement to provide all the information necessary for researchers to access this dataset (eg, accession number, url or email address to contact if an application is required).

5- Please add a bit more context of why the study is important to the Abstract Background (i.e. note the limited availability of same-day microbiological testing an the risk of losing patients to follow-up). 

6- In the Abstract Methods and Findings, please include the dates of the two cohorts, and some relevant demographic information (e.g., age, sex, HIV positivity).

7- The sentence on lines 44-46 needs a bit more context for what the scores of 3 and 7 mean.

8- Thank you for adding the limitations of the score on lines 46-47, please also include limitations of the study itself, though.

9- Similar to the Abstract Background section, the Abstract Conclusions could use a bit more context as well. Please address the study implications without overreaching what can be concluded from the data; the phrase "In this study, we observed ..." may be useful. 

10- Around line 51, you could add "subject to prospective validation" or similar

11- Please move the reference brackets in front of punctuation. E.g.: “... the leading single-agent cause of infectious mortality worldwide [1].”

12- Table 1: please make clear that the numbers in the parentheses are percentages. Also, “irregular” is misspelt.

13- Table 2: please include p-values alongside the 95% CIs.

14- The numbers above the x-axis on figure 4 are a bit difficult to interpret. Please consider reformatting the “Score” and “Post-test probability” labels with smaller fonts and re-aligning. Perhaps the scales below could be smaller to allow more space?

15- Please use the "Vancouver" style for reference formatting, and see our website for other reference guidelines https://journals.plos.org/plosmedicine/s/submission-guidelines#loc-references

> Reference 7, 14, 15 may need reformatting. 

> For reference 17, 18, please provide the date accessed.

> Missing information from Reference 31?

16- Figure S1 could be moved to the main text. The background renders in dark grey on my computer, which makes the arrows difficult to see.

17- The third image from Figure S2 may need to be edited to blur the face of the person on the right. I think he’s probably identifiable.

18- References to the “Appendix” throughout the the main text should be changed to reference the relevant supplementary file(s).

19- Please remove the funding information from the "Acknowledgements" at the end.

Comments from Reviewers:

Reviewer #2: No remaining requests.

Reviewer #3: I'm happy with the responses to the reviewer comments and revisions to the paper. 

Reviewer #4: We thank the authors for addressing most of our points from the first review round, particularly the requested informative additions to Figure 4 (former Figure 2):

3. The expected sensitivity/specificity for Xpert (0.88/0.98) is now provided.

4. Single imputation values might be derived from Table 1 on hindsight, which might or might not be further emphasized.

5. WHO TB symptoms were briefly expounded on.

6. Symptoms > 14 days was clarified as being any TB symptoms.

7. Table 2 Footnote c was clarified by more fully describing a method to translate the LASSO coefficients to actionable scores.

8. The cost-benefit analysis of Figure 4 (former Figure 2) has been more quantitatively justified.

9. Sample patient scores were labeled on Figure 4 (former Figure 2) as requested.

10. Reported predictive accuracies of various models were mentioned as c-statistics.

11. The limitation on community-based triaged has been emphasized as suggested.

12. Additional caveats have been described discussion appropriate adjustments, were microbiological results not as expected according to the assumed TB probabilities. Ideally, the effect of incorrect TB probability assumptions might be more quantitatively explored.

[LINK]

---

## [Editor Report · Decision Letter 3]

14 Oct 2020

Dear Dr Baik, 

On behalf of my colleagues and the academic editor, Dr. Ankur Gupta-Wright, I am delighted to inform you that your manuscript entitled "A clinical score for identifying active tuberculosis while awaiting microbiological results: development and validation of a multivariable prediction model in sub-Saharan Africa" (PMEDICINE-D-20-01816R3) has been accepted for publication in PLOS Medicine. 

PRODUCTION PROCESS

Before publication you will see the copyedited word document (within 5 busines days) and a PDF proof shortly after that. The copyeditor will be in touch shortly before sending you the copyedited Word document. We will make some revisions at copyediting stage to conform to our general style, and for clarification. When you receive this version you should check and revise it very carefully, including figures, tables, references, and supporting information, because corrections at the next stage (proofs) will be strictly limited to (1) errors in author names or affiliations, (2) errors of scientific fact that would cause misunderstandings to readers, and (3) printer's (introduced) errors. Please return the copyedited file within 2 business days in order to ensure timely delivery of the PDF proof. 

If you are likely to be away when either this document or the proof is sent, please ensure we have contact information of a second person, as we will need you to respond quickly at each point. Given the disruptions resulting from the ongoing COVID-19 pandemic, there may be delays in the production process. We apologise in advance for any inconvenience caused and will do our best to minimize impact as far as possible.

PRESS

PROFILE INFORMATION

Thank you again for submitting the manuscript to PLOS Medicine. We look forward to publishing it. 

Best wishes, 

Thomas McBride, PhD

Senior Editor 

PLOS Medicine

plosmedicine.org